# Characteristics and Geological Significance of CO$_2$-Rich Fluid Inclusions in Dakalasu No. 1 Pegmatite Dyke, Altay

**Jiehao Zhou** [1], **Hui Zhang** [2], **Yong Tang** [2], **Zhenghang Lv** [2] **and Shenjin Guan** [1,*]

1 Faculty of Land Resource Engineering, Kunming University of Science and Technology, Kunming 650031, China
2 Key Laboratory of High-Temperature and High-Pressure Study of the Earth's Interior, Institute of Geochemistry, Chinese Academy Sciences, Guiyang 550081, China
* Correspondence: guansj@kust.edu.cn

**Abstract:** The fluids in of pegmatite rare metal deposits are generally rich in rare metal elements and volatiles (B, P, F, H$_2$O, CO$_2$, etc.), and they have a high capacity for dissolving and migrating rare metals. The Dakalasu No. 1 rare metal pegmatite vein is located in northwest China's Altay orogenic belt. Previous studies have indicated that it is a small- to medium-sized beryllium-niobium-tantalum deposit. It showed significant mineral assemblage zonations from the rim to the core, and the mineralizing fluids define a volatile-rich NaCl-H$_2$O-CO$_2$ $\pm$ CH$_4$ system. In this contribution, beryl and quartz, which are widely developed in each mineral association and textural zone, were selected for fluid inclusion research through detailed petrographic investigation, microthermometry, and LA-ICP-MS analysis. Petrographic results show that at least three types of fluid inclusions are developed in each mineral textural zone. They are CO$_2$-rich inclusions (type I), gas-liquid two-phase inclusions (type II), and daughter mineral-bearing inclusions (type III), respectively. Additionally, minor melt inclusions (type IV) are visible in the beryl from the rim zone. Microthermometric measurements showed that the homogenization temperature of fluid inclusions in the rim zone was concentrated between 242 °C and 293 °C, with an average of 267 °C, and the salinity was between 7.2–10.3 wt% NaCl$_{eqv}$, with an average of 8.6 wt% NaCl$_{eqv}$. In comparison, the temperature of the core zone was in the range of 225–278 °C, with an average of 246 °C, and the salinity focused between 6.0–7.7 wt% NaCl$_{eqv}$, with an average of 7.1 wt% NaCl$_{eqv}$. The quantitative analysis of individual inclusions by LA-ICP-MS revealed that Li, B, K, Zn, Rb, Sb, Cs, and As were relatively enriched in the rim zone. In contrast, the core zone showed a decreasing trend in trace elements such as Li, B, K, Rb, and Cs. The CO$_2$ content in the fluid exhibited the same decreasing trend from the rim to the core zone, indicating that volatile components such as CO$_2$ played an essential role in the migration and enrichment of rare metal elements. The melt-fluid immiscibility is likely to be a necessary mechanism for significantly enriching rare metals in the Dakalasu No. 1 pegmatite dyke.

**Keywords:** CO$_2$-rich fluid; LA-ICP-MS; enrichment mechanism; pegmatite-type rare metal deposits; Dakalasu; Altay





## 1. Introduction

Rare metals such as Li, Be, Nb, Ta, Rb, and Cs played an irreplaceable role in aerospace and defense science and technology. Due to their scarcity, they are listed as strategic essential mineral resources by China, Europe, and the United States [1–4]. Granitic pegmatite is essential for rare metal deposits. Researchers have highly valued its metallogenic theory and prospecting work [3,5–13].

The Altay Orogenic Belt (AOB) is located in the northern part of Xinjiang, China. It is an essential part of the Central Asian Orogenic Belt (CAOB). More than 100,000 pegmatite veins are distributed in about 20,000 square kilometers [14], of which more than 95% are concentrated in 38 pegmatite ore fields [15]. Most pegmatite dykes host rare metal

mineralization and significant mineral textural zonation. Therefore, the AOB is a natural laboratory for studying the formation mechanism of pegmatite-type rare metal deposits.

Previous studies have shown that there are four stages of rare metal mineralization in the AOB, namely Devonian-Early Carboniferous (403~333 Ma), Permian (275~250 Ma), Triassic (248~200 Ma), and Jurassic (199~157 Ma), with Permian and Triassic being the most active. [16–23]. Pegmatite-type rare metal deposits with different scales have been formed. The Koktokay No.3 pegmatite dyke is an ultra-large Li-Be-Nb-Ta-Rb-Cs deposit that is famous across the globe, and Kaluan is a large to ultra-large Li deposit. Since the 1980s, researchers from China have conducted much research on pegmatites' mineral composition, mineralization characteristics, genesis and evolution, and rare metal resource evaluation [24–34]. Fruitful results have been obtained, laying a solid foundation for the research on the genesis of granitic pegmatites and the mechanism of rare metal mineralization.

Be, Nb, and Ta mineralization characterize the Dakalasu No. 1 granitic pegmatite in the AOB. It is located about 36 km southeast of Altay City, Xinjiang, with medium-sized niobium-tantalum and small-sized beryllium reserves [35]. The chronological results show that it formed in a Permian post-orogenic extensional environment [36,37]. Recent studies on beryl mineralogy and fluid inclusions suggest that the ore-forming fluids are characterized by high $CO_2$ enrichment. Still, the characteristics of the mineralized fluids and the enrichment mechanism of rare metal elements remain debatable [38].

LA-ICP-MS analysis of the composition of individual inclusions has the characteristics of high precision, low detection limit, and multi-element micro-area detection. It has incomparable advantages over traditional methods in exploring the geochemical properties of ore-forming fluids and revealing the metallogenic mechanism [39–45]. Through detailed field research and petrographic studies, combined with microthermometry, Laser Raman spectroscopy, and LA-ICP-MS, this contribution investigates the fluid inclusions captured by beryl and quartz in various mineral textural zone of the Dakalasu No. 1 pegmatite dyke. The fluid nature of this deposit and the role of volatile components such as $CO_2$ on rare metal enrichment mineralization are further examined.

## 2. Geological Background

The AOB is located between the Siberian and the Kazakhstan-Zhungeer plate. The overall length of the orogenic belt is about 500 km and the width is about 40–80 km. It is a Phanerozoic accretionary orogenic belt with the characteristics of multi-continental blocks, island arcs, and accretionary complex belts [46–50]. It has undergone a complex and extensive tectonic evolution process, from the formation of ancient continental blocks, the proliferation of continental crust, and the separation of plates to the final aggregation into a unified and stable continent [51–53]. According to the characteristics of metamorphism and faults, the Altay area was divided into four terranes by the Hongshanzui-Nort, Kurt-Abba Palace, and Ertix faults [54,55]. From north to south, these are North Altay, Central Altay, Qiongkuer, and South Altay terranes (Figure 1).

The Dakalasu mining area is part of the Dakalasu-Kekexier pegmatite field in the Dakalasu-Jiamanbaha rare metal metallogenic subbelt, located about 36 km southeast of Altay City, Xinjiang. The exposed strata in the area are Middle-Upper Devonian schist, gneiss, and metamorphic siltstone, Late Hercynian porphyritic biotite granite, and micaceous granite (Figure 2) [56]. More than ten pegmatite veins with rare metal mineralization, represented by the Dakalasu No. 1 pegmatite vein, are densely intruded into the southern part of the Dakalasu granite body.

From the rim to the core, the Dakalasu No. 1 pegmatite vein is mainly divided into four mineral textural zones, namely graphic pegmatite, blocky microcline, quartz-muscovite, and quartz core (Figure 3). Tourmaline is widely developed in the pegmatite/granite contact zone. The mineral assemblage in the graphic texture zone mainly comprises quartz, albite, beryl, and tourmaline. The blocky microcline zone is mainly composed of megacryst blocky microcline. The mineral composition of the quartz-muscovite zone consists of quartz, muscovite, and beryl, and the core zone is primarily composed of quartz

(Figure 4A,B,D,E). The main beryllium-bearing minerals are yellow-green and golden-yellow beryl. The beryl content within the central blocky pegmatite zone is as high as 0.49 vol% and contains considerable amounts of coarse crystalline beryl. The content of beryl in the whole vein is of about 0.56 vol%, the vein also containing 0.022 vol% tantalum- and niobium-bearing minerals [35]. Tantalum- and niobium-bearing minerals include columbite, strueverite, tapiolite, and apatite. The medium-coarse-grained pegmatite at the rim zone mainly contains columbite and strueverite, while tapiolite and microlite are primarily the core blocky pegmatite zone. Among the 38 pegmatite fields in Altay, this zoning feature of niobium-tantalum minerals can be observed only in the Dakalsu-Kekexier and the nearby Abagong pegmatite field. The Daikalasu No. 1 pegmatite vein is the most representative [35].

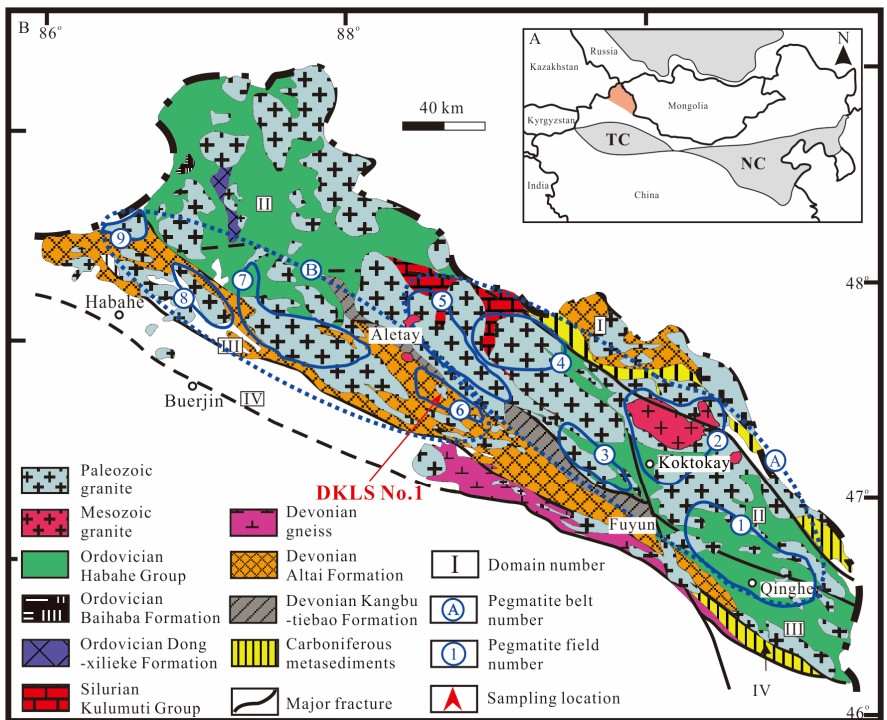

**Figure 1.** Regional geological map of the Chinese Altay, modified after [35,57,58]. (**A**) Main tectonic units of the Chinese Altay orogen and adjacent areas; (**B**) Geological map of the Chinese Altay, showing the location of mineralized pegmatites and deposits.

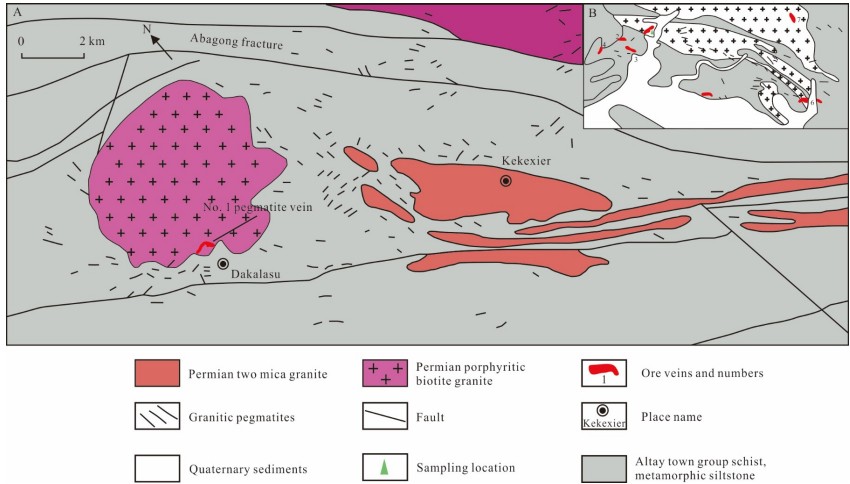

**Figure 2.** Geological sketch of the Dakalasu pegmatite veins in the Altay region of Xinjiang, modified after [35,56]. (**A**) Geological overview of Dakalasu pegmatite ore cluster; (**B**) Simplified Geological Map of Dakalasu Be-Nb-Ta Ore District.

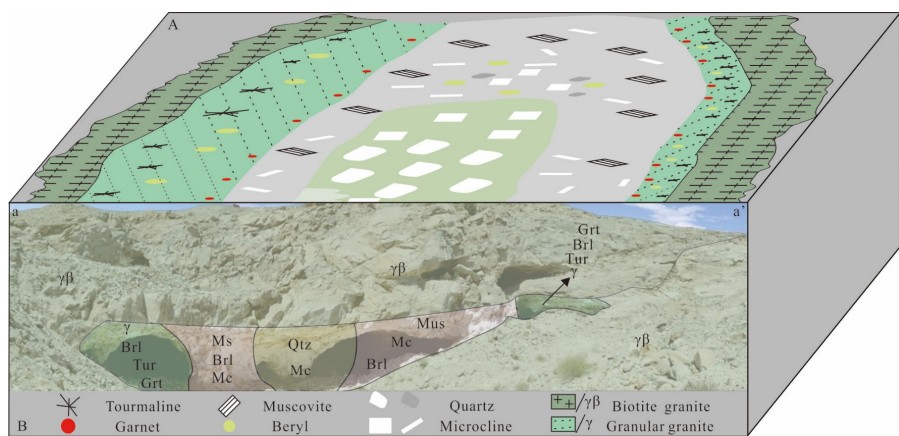

**Figure 3.** Schematic map of the mineral textural zonation of Dakalasu No. 1 pegmatite dyke, modified after [38]. (**A**) The layout of the Dakarasu No. 1 pegmatite vein zonation; (**B**) Dakalasu No.1 pegmatite vein zoning profile.

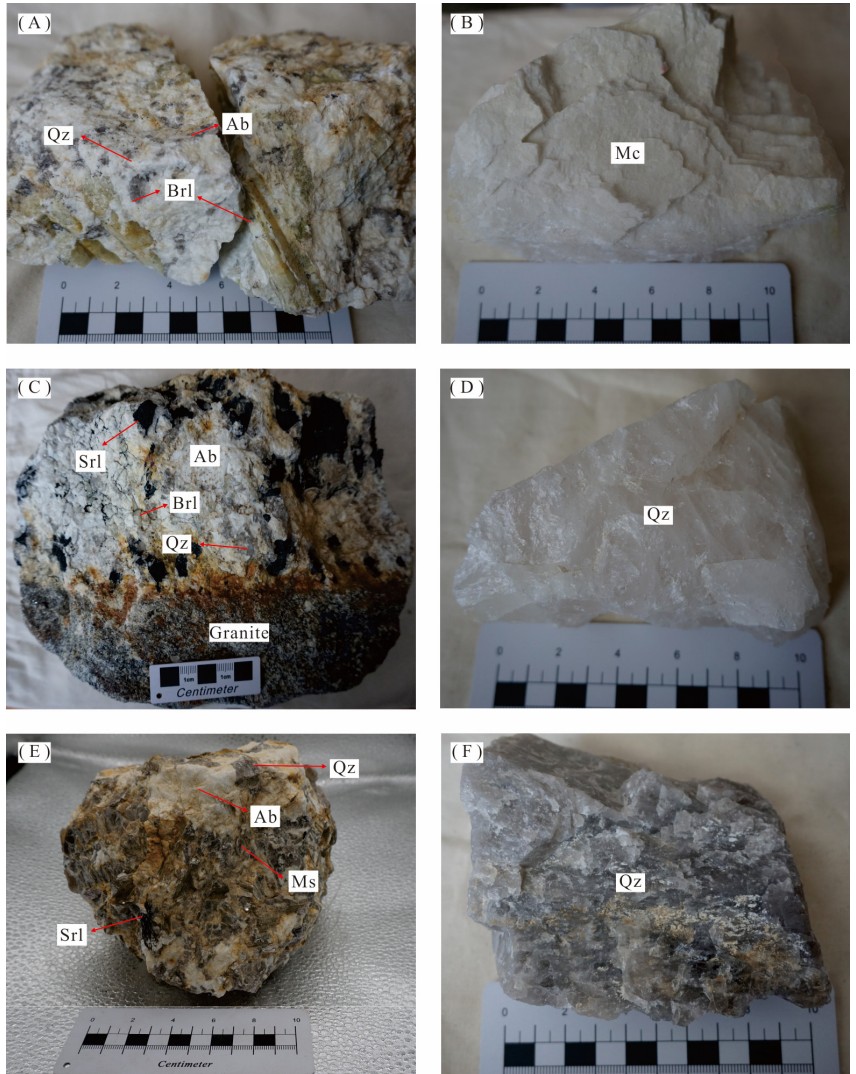

**Figure 4.** Photographs of ores from the Dakalasu No. 1 pegmatite vein. (**A**) Graphic texture zone; (**B**) Block microcline feldspar; (**C**) Contact zone between biotite granite and graphic texture pegmatite; (**D**) Rose quartz from the transitional zone; (**E**) Quartz—muscovite zone; (**F**) Quartz from the quartz core. Quartz = Qz; Albite = Ab; Microcline = Mc; Schorl = Srl; Beryl = Brl; Muscovite = Ms.

## 3. Sampling and Methodology

The samples used in this study were all taken from the open pit of the Dakalasu No. 1 pegmatite vein. Among them, seven samples are from the graphic texture zone (rim), one piece is from the quartz-muscovite zone (transition), and four are from the quartz core (Figure 4D–F). The samples were cut into thin sections (~0.2 mm), polished on both sides, and petrographically examined under an optical microscope (Figures 5 and 6).

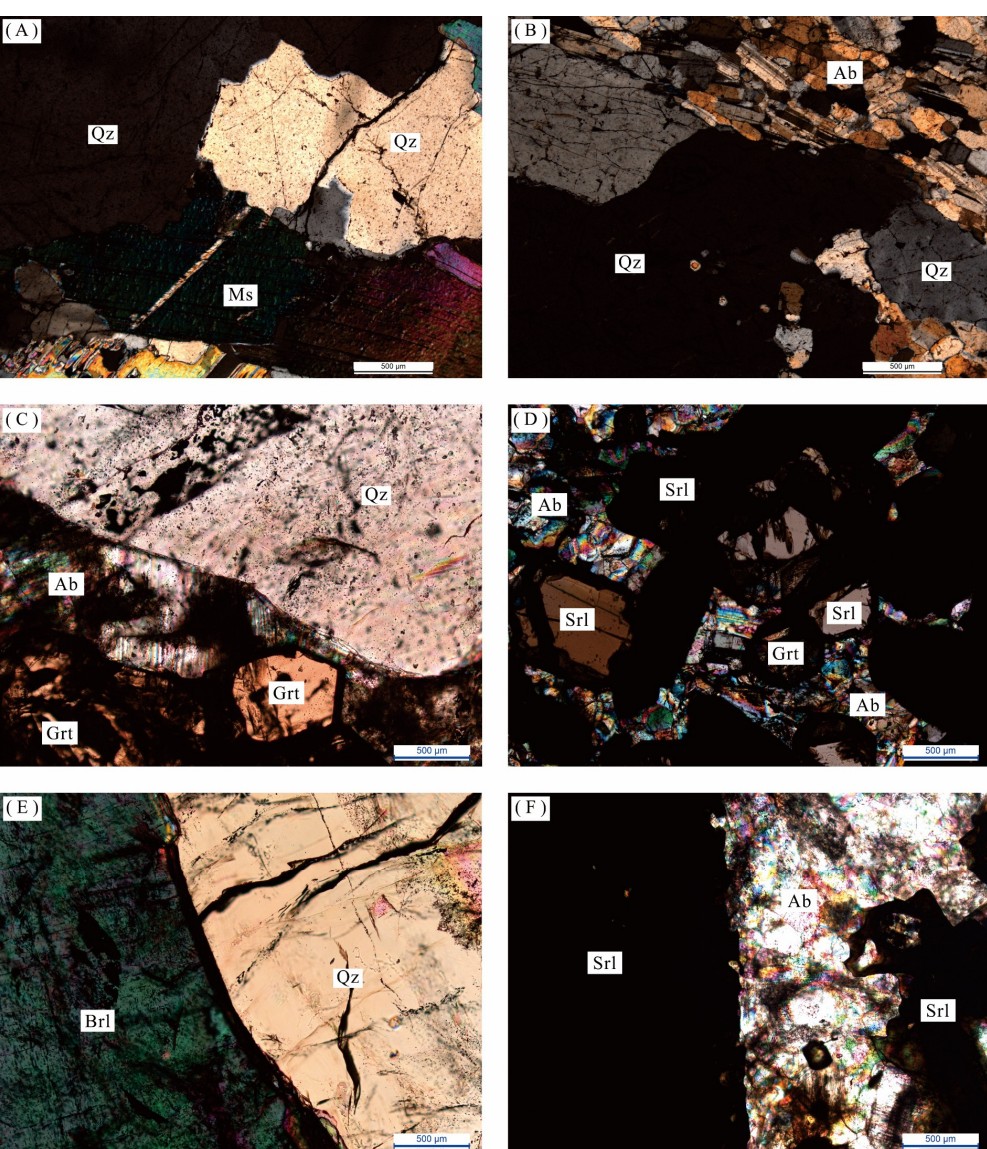

**Figure 5.** Microscopic characteristics of samples from different mineral textural zones. (**A**) Quartz-muscovite combination (crossed polarized light); (**B**) Quartz-albitite combination (crossed polarized light); (**C**) Quartz-albite-garnet combination (crossed polarized light); (**D**) Albite-schorl-garnet combination (crossed polarized light); (**E**) Quartz-beryl combination (crossed polarized light); (**F**) Albite-schorl combination (crossed polarized light). Quartz = Qz; Albite = Ab; Microcline = Mc; Schorl = Srl; Beryl = Brl; Muscovite = Ms; Garnet = Grt.

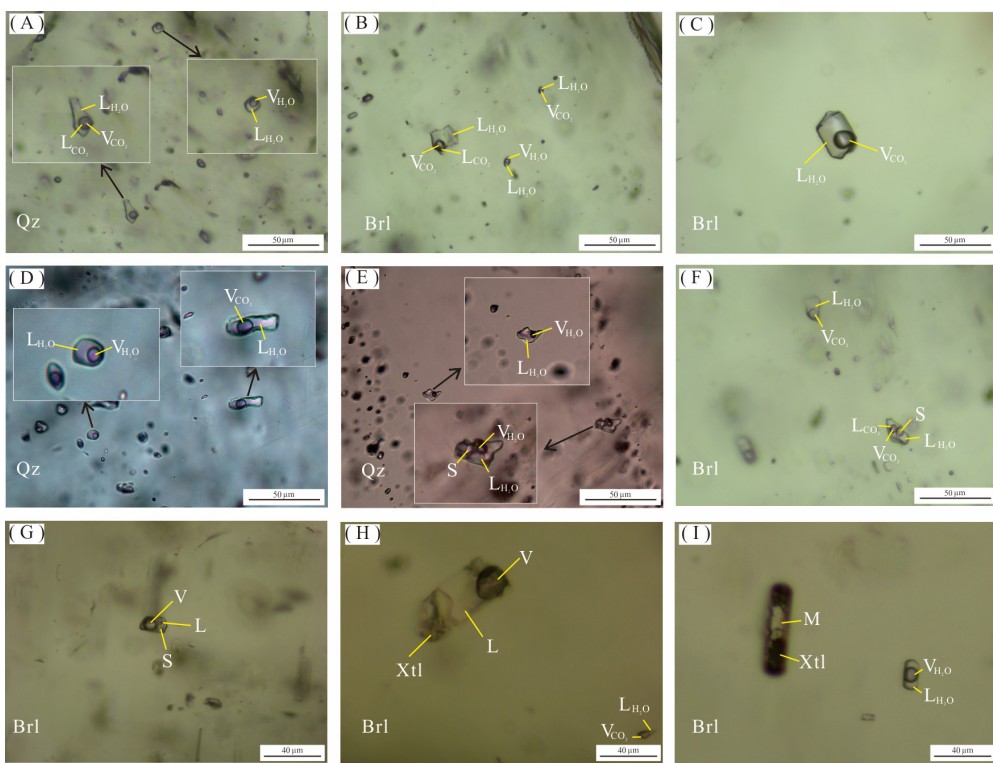

**Figure 6.** Microscopic photos of fluid and melt inclusions of the Dakalasu No. 1 pegmatite vein. (**A**,**B**) Three-phase $CO_2$ inclusions and two-phase gas-liquid inclusions in quartz and beryl; (**C**) $CO_2$-bearing two-phase aqueous inclusions in beryl; (**D**) Gas-liquid two-phase inclusions in quartz and $CO_2$-bearing two-phase aqueous inclusions; (**E**–**G**) Daughter mineral-bearing inclusions and gas-liquid two-phase inclusions in quartz and beryl; (**H**) Fluid-melt inclusions coexist with $CO_2$-bearing two-phase aqueous inclusions in beryl; (**I**) Crystalline silicate melt inclusions coexist with gas-liquid two-phase inclusions in beryl.

### 3.1. Microthermometry

Microthermetric measurement of fluid inclusions was completed at the Fluid Inclusion Laboratory of the School of Land and Resource Engineering, Kunming University of Science and Technology. The instrument used was the British Linkam THMSG-600 heating-cooling stage with a temperature range of −196 °C to 600 °C. The stage was calibrated to an uncertainty of ± 0.2 °C in the field of −56.6 to 0.0 °C. At temperatures above 100 °C, the precision is ± 2 °C. The salinities of NaCl-$H_2O$ inclusions were calculated using the final melting temperatures of ice or dissolution temperature of halite using the equations of Bodnar (1993) [59]. The salinities of $CO_2$-bearing fluid inclusions were calculated using the melting temperatures of clathrate [60,61].

### 3.2. Raman Spectroscopy

Gas compositions of fluid inclusions were identified with a Renishaw InVia Reflex Raman microprobe in the same laboratory. An Ar-ion laser with a surface power of 5 mW was used for exciting the radiation (514.5 nm), the area of the charge-coupled device (CCD) detector is 20 $\mu m^2$, and the scanning range for spectra was set between 100 and 4000 $cm^{-1}$, with an accumulation time of 30 s for each scan.

### 3.3. LA-ICP-MS Microanalysis

Laser-ablation ICP-MS (Agilent Technologies: Santa Clara, CA, USA) was used to quantify the compositions of single fluid inclusions at the State Key Laboratory for Mineral Deposit Research, Nanjing University. The measurement was carried out using a Coherent GeolasHD system equipped with a 193 nm ArF laser coupled to a NexION 350 ICP mass

spectrometer. NIST 610 was used as an external standard for all elements and instrumental drift correction. The NaCl equivalent salinities obtained from microthermometry were used as internal standards [62,63]. Major cations such as Na and K were included in the salt correction. Data reduction was performed using the SILLS software [64]. Average elemental concentrations with $\pm 1\sigma$ uncertainty were calculated for each fluid inclusion assemblage. For detailed measurement procedures, see [65].

## 4. Results

### 4.1. Fluid Inclusion Petrography

Numerous fluid inclusions, including primary, secondary, and pseudo-secondary, are developed in beryl and quartz in each textural zone of the Dakalasu No. 1 pegmatite vein. Primary inclusions were selected for the focused study. According to the composition of fluid inclusions and their phase properties at room temperature [66], fluid inclusions are classified into four types, namely, $CO_2$-bearing inclusions (type I), gas-liquid phase two-phase inclusions (type II), daughter mineral-bearing inclusions (type III), and melt inclusions (type IV).

The type $I_a$ $CO_2$-bearing fluid inclusion is widely distributed in a variety of minerals in both the graphic pegmatite zone and the quartz core and can be further divided into two subclasses, $I_a$ and $I_b$. Type $I_a$ inclusions are about 12–40 μm in size, primarily distributed in clusters, a few isolated, with irregular, ellipsoidal, and negative crystal shape morphology. Three phases are visible at room temperature (20 °C), i.e., liquid $H_2O$, liquid $CO_2$, and gas phase $CO_2$ (Figure 6A). Most inclusions decrepitated during the microthermometric measurement, indicating a high internal pressure. The size and shape of type $I_b$ inclusions are similar to those of $I_a$, but only two phases are visible at room temperature, i.e., liquid water and gas-phase $CO_2$. In the Raman spectroscopy analysis, 1285 $cm^{-1}$ and 1388 $cm^{-1}$ $CO_2$ characteristic peaks were visible in the liquid phase.

Type II is gas-liquid two-phase inclusions, mainly distributed in quartz in the graphic texture zone and quartz core (Figure 6D,E). This type of inclusion is less in number and usually smaller in size (~10 μm). Most are isolated and irregular in shape. The vapor phase accounts for about 10%–30% of the volume of the inclusions.

Type III, daughter mineral-bearing inclusions, is mainly developed in beryl and quartz in the graphic textural zone. The number of this type of inclusion is small, and the morphology is generally irregular. It typically comprises three phases: liquid water, gaseous phase, and daughter minerals. The shape of the daughter minerals is generally cubic or spherical, these minerals being most probable Na or K chlorides.

Type IV, melt inclusions, is mainly developed in beryl of the graphic textural zone. The number of inclusions of this type is small. It can be subdivided into two categories, water-rich melt inclusions and water-poor melt inclusions, according to the content of the fluid. The water-rich melt inclusions are only isolated in beryl, with a size of about 25 μm and mostly negative crystal with columnar shape morphology. It is mainly composed of liquid water, vapor bubble, and crystalline silicates, and the liquid phase accounts for about 40%–50% of the volume of the inclusions (Figure 6H). The water-poor melt inclusions also occur only in beryl, with a size of about 20 μm and a negative crystal shape or elongated morphology. It consists of glassy melt and vapor bubbles. They usually coexist with gas-liquid two-phase inclusions (Figure 6I).

### 4.2. Microthermometry

The microthermometric measurement was mainly focused on type I and type II inclusions, with the host minerals, including beryl, rose quartz, and quartz, widely distributed in the rim, transition, and core. The results are shown in Table 1 and Figures 7 and 8.

**Table 1.** Microthermometry results of the fluid inclusions from Dakalasu No. 1 pegmatite vein.

| Stage Division | Inclusion Type | Total Number | Size (μm) | $T_{m, CO_2}$ (°C) | $T_{m, ice}$ (°C) | $T_{m, clath}$ (°C) | $T_{h, CO_2}$ (°C) | $T_{h, tot}$ (°C) | Salinity (wt% NaCl$_{eqv}$) | CO$_2$ Phase Density (g/cm$^3$) |
|---|---|---|---|---|---|---|---|---|---|---|
| rim zone (beryl) | I$_a$ | 26 | 5~30 | −58.1~−58.6 | | 4.1~5.7 | 13.5~25.6 | 242.5~293.4 | 8.0~10.3 | 0.71~0.84 |
| | I$_b$ | 18 | 5~22 | −58.2~−58.5 | | 4.7~6.1 | | 229.7~280.8 | 7.2~9.4 | |
| | II | 6 | 8~19 | | −6.2~−3.9 | | | 231.3~242.7 | 6.3~9.47 | |
| transition zone (rose quartz) | I$_a$ | 16 | 6~23 | −58.2~−58.4 | | 5.0~7.0 | 24.1~26.5 | 253.4~287.5 | 5.7~9 | 0.69~0.73 |
| | I$_b$ | 12 | 7~20 | −58.1~−58.4 | | 5.2~6.5 | | 240.8~299.4 | 6.5~8.6 | |
| | II | 9 | 8~17 | | −6.9~−4.2 | | | 240.8~283.8 | 6.7~10.4 | |
| quartz core (quartz) | I$_a$ | 29 | 7~29 | −58.2~−58.4 | | 5.1~6.8 | 19.5~28.7 | 228.7~278.3 | 6.0~7.7 | 0.65~0.78 |
| | I$_b$ | 12 | 8~18 | −58.1~−58.4 | | 5.2~6.8 | | 225.5~262.5 | 6.0~7.7 | |
| | II | 7 | 6~15 | | −7~−4.4 | | | 232.2~267.5 | 7.02~10.5 | |

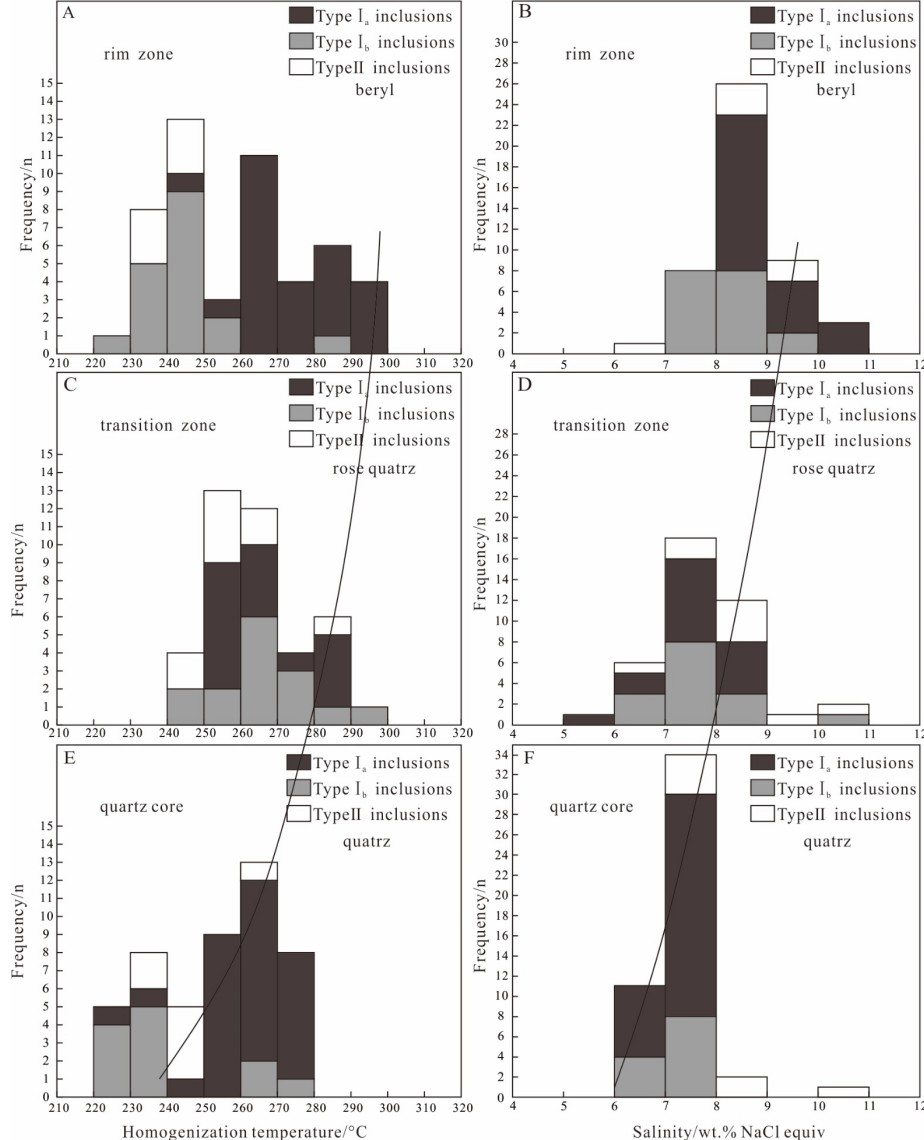

**Figure 7.** Homogeneous temperature and salinity histogram of the fluid inclusions from Dakalasu No. 1 pegmatite vein. (**A**,**B**) Histogram of homogenization temperature and salinity of fluid inclusions in beryl; (**C**,**D**) Histogram of homogenization temperature and salinity of fluid inclusions in in rose quartz; (**E**,**F**) Histogram of homogenization temperature and salinity of fluid inclusions in quartz.

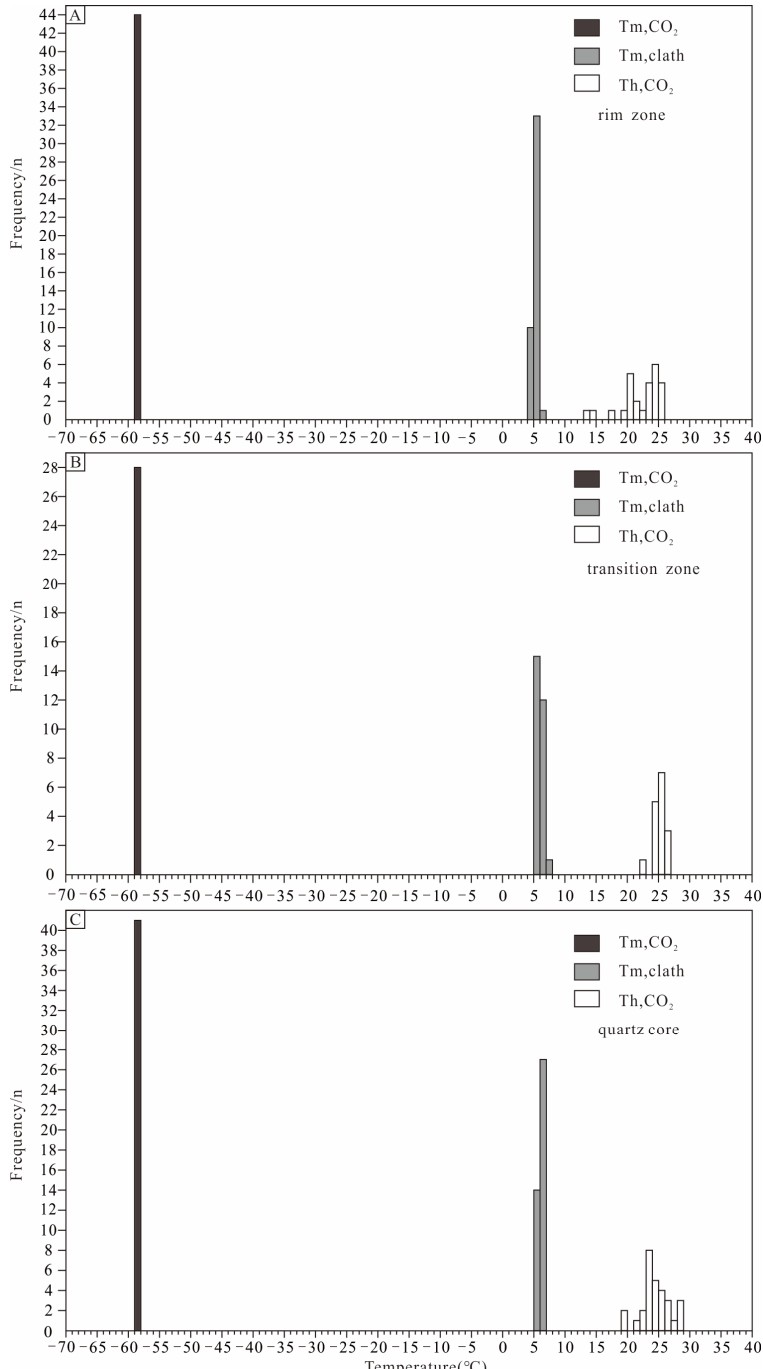

**Figure 8.** Histograms of $CO_2$ melting temperatures, clathrate melting temperatures, and $CO_2$ partial homogenization temperatures of type $I_a$ and type $I_b$ inclusions in Dakalasu No. 1 pegmatite vein. (**A**) Type $I_a$ and type $I_b$ inclusions in the rim zone; (**B**) type $I_a$ and type $I_b$ inclusions in the transition belt; (**C**) type $I_a$ and type $I_b$ inclusions in the quartz belt.

In the beryl of the rim zone, the $CO_2$ triple-phase point temperature ($T_{m, CO_2}$) of type I inclusions was between $-58.1$ and $-58.6$ °C, indicating that its gas phase is not pure $CO_2$. This is consistent with the results of Raman spectroscopy, where the gas phase composition contains moderate amounts of $CH_4$ and $N_2$. During slow heating, the melting temperature of $CO_2$ clathrate ($T_{m, clath}$) was observed between 4.1 and 5.7 °C. Salinities of 8.0%~10.3% $NaCl_{eqv}$ were calculated according to [60]. The partially homogeneous temperature of the $CO_2$ phase ($T_{h, CO_2}$) ranges from 19.5 to 25.6 °C. During continuous heating, the inclusions gradually homogenize to the liquid phase, and the completely homogeneous temperature

ranges from 243 to 294 °C. According to [67] and [68], the $CO_2$ phase density was calculated to be 0.71–0.84 g/cm$^3$. For type II inclusions in beryl, the freezing point was measured in the range of −6.2 to −3.9 °C during reheating after complete freezing. According to [59], the calculated salinity ranged from 6.3 to 9.5% NaCl$_{eqv}$. The inclusions were finally homogenized to the liquid phase with a temperature between 232 and 243 °C.

In the rose quartz of the transition zone, the $CO_2$ triple-phase point temperature of type I inclusions ranges from −58.2 to −58.4 °C, similar to that in beryl. The melting temperature of $CO_2$ clathrate varies from 5.0 to 7.0 °C, corresponding to a salinity of 5.7 to 9% NaCleqv. The $CO_2$ partial homogeneity temperature is between 24.1 and 26.5 °C, and the complete homogeneity temperature ranges from 253 to 288 °C. The $CO_2$ phase density is 0.69 to 0.73 g/cm$^3$, slightly lower than that in beryl. The type II inclusions in the rose quartz exhibit a freezing point of −6.9 to −4.2 °C, corresponding to a salinity of 6.7 to 10.4% NaCl$_{eqv}$. The homogeneous temperature lies between 241 and 284 °C.

In the quartz of the core zone, the $CO_2$ triple-phase point temperature of type I inclusions also falls between −58.2 and −58.4 °C. The $CO_2$ clathrate disappears at about 5.1–6.8 °C, corresponding to a salinity of 6.0%–7.7% NaCl$_{eqv}$. The $CO_2$ phase is partially homogeneous at 18.8~28.7 °C and completely homogeneous to the liquid phase at 229~278 °C. The calculated $CO_2$ phase density is from 0.65 to 0.78 g/cm$^3$, similar to or slightly lower than rose quartz. The freezing point of type II inclusions in quartz ranges from −7 to −4.4 °C, corresponding to a salinity of 7.0 to 10.5% NaCl$_{eqv}$. The inclusions finally homogenize to the liquid phase at 232 to 267 °C.

*4.3. Raman Spectroscopy*

The results of Raman spectroscopy showed that the gas phase of type I$_a$ inclusions contained significant $CO_2$ characteristic peaks and modest levels of $N_2$ and $CH_4$; $CO_2$ was detected in both the liquid and gas phases (Figure 9A,B). Two characteristic peaks of $CO_2$ (1285 cm$^{-1}$, 1387 cm$^{-1}$) were clearly seen in the type I$_b$ inclusions' vapor and liquid phase, but $N_2$ and $CH_4$ were not recognized in the gas phase (Figure 9C,D). At 3607 cm$^{-1}$, the characteristic peak of water in beryl crystal structure can be seen [69].

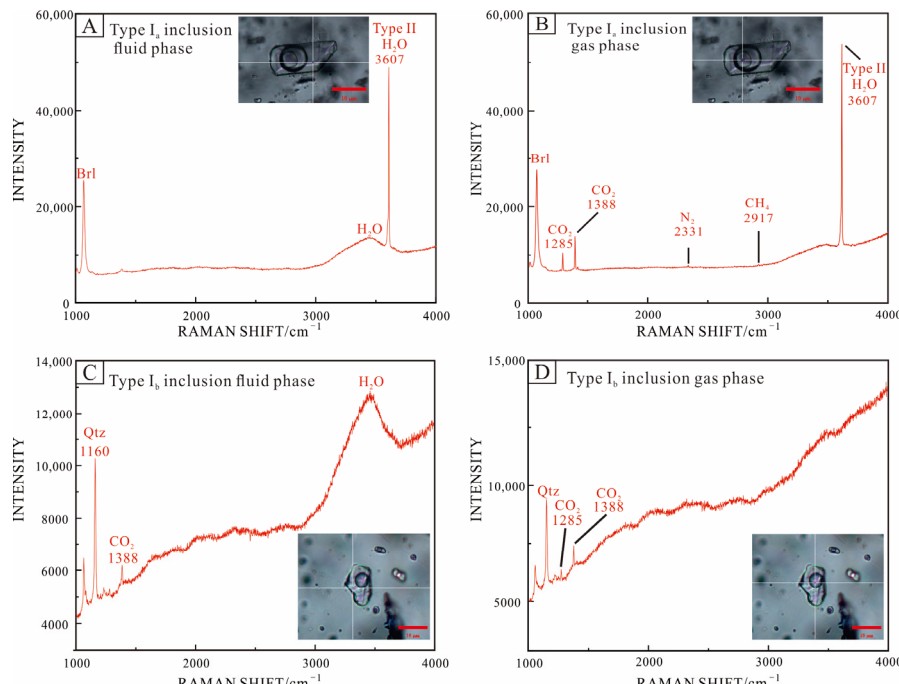

**Figure 9.** Raman analysis of representative fluid inclusions in beryl (**A**,**B**) and quartz (**C**,**D**) from the Dakalasu No. 1 pegmatite vein. (**A**,**B**) Three-phase $CO_2$ inclusions (type I$_a$); (**C**,**D**) $CO_2$-bearing aqueous inclusions (type I$_b$).

### 4.4. LA-ICP-MS

In this contribution, 22 $CO_2$-bearing fluid inclusions in beryl, quartz, and rose quartz of Dakalasu No. 1 pegmatite dyke were selected for LA-ICP-MS analysis. The results are shown in Tables 2–4 and Figure 10.

**Table 2.** LA-ICP-MS in situ analysis of fluid inclusions in beryl of Dakalasu No. 1 pegmatite vein (ppm).

| Sample | A32 | A33 | A34 | A35 | A36 | A41 | A43 |
|---|---|---|---|---|---|---|---|
| Li | 737.7 | 614.8 | 527.1 | 2172 | 316.5 | 390.3 | 1659 |
| B | 2889 | 5702 | 976.7 | 1924 | 3525 | 34,407 | 10,648 |
| Na | 35,917 | 33,478 | 39,497 | 33,478 | 33,478 | 35,917 | 33,478 |
| K | 2813 | 3281 | 2932 | 641.2 | 4052 | 8880 | 3737 |
| Zn | 37.16 | 474.4 | 151.3 | - | 314.1 | 891.1 | - |
| As | 684.3 | 1143 | 120.8 | 194.2 | 171.7 | - | 2735 |
| Rb | 98.46 | 118.9 | 116.7 | 54.74 | 88.15 | 265.2 | 322.9 |
| Sr | 1.40 | 11.59 | 32.28 | - | - | - | - |
| Sb | 211.4 | 139.7 | 120.2 | - | 233.4 | 1717 | 855.6 |
| Cs | 1005 | 1646 | 669.6 | 1505 | 1412 | 1420 | 3842 |
| Ta | 1.23 | - | 1.21 | - | - | - | - |
| W | - | - | 178.8 | - | 43.67 | - | 277.4 |
| Rb/Na | 0.002741 | 0.003552 | 0.002955 | 0.001635 | 0.002633 | 0.007384 | 0.009646 |
| Cs/Na | 0.02799 | 0.04917 | 0.01695 | 0.04497 | 0.04216 | 0.03954 | 0.1148 |
| Cs/Rb | 10.21 | 13.84 | 5.737 | 27.50 | 16.01 | 5.355 | 11.90 |

Notes: "-" means below detection limit.

**Table 3.** LA-ICP-MS in situ analysis of fluid inclusions in rose quartz of Dakalasu No. 1 pegmatite vein (ppm).

| Sample | A05 | A06 | A06-2 | A07 | A08 | A09 | A11 | A12 |
|---|---|---|---|---|---|---|---|---|
| Li | 1259 | 1827 | 1186 | 3973 | - | 820.0 | 1353 | 4009 |
| Be | - | - | | - | - | 217.5 | - | - |
| B | 11,981 | 13,446 | 12,314 | 2994 | 2326 | 11,767 | 10,988 | 11,130 |
| Na | 27,066 | 30,291 | 30,291 | 22,345 | 25,728 | 27,734 | 28,364 | 22,345 |
| K | - | 679.4 | 1424 | 2701 | 2580 | 241.8 | 734.9 | - |
| Ti | - | - | - | 1496 | 692.6 | 3468 | 334.9 | 4655 |
| As | 2939 | 840.9 | 1593 | 3281 | 1134 | 4468 | 2710 | - |
| Rb | 15.19 | 16.85 | - | 27.74 | 34.32 | 8.06 | 10.30 | - |
| Sb | - | - | - | 137.04 | - | - | 18.76 | - |
| Cs | 551.9 | 646.2 | 602.9 | 931.9 | 461.2 | 202.4 | 234.2 | 332.2 |
| Rb/Na | 0.000561 | 0.000556 | - | 0.001241 | 0.001334 | 0.000291 | 0.000363 | - |
| Cs/Na | 0.02133 | 0.01991 | 0.03368 | - | 0.01793 | 0.007297 | 0.008259 | 0.001487 |
| Cs/Rb | 36.29 | 38.35 | - | 33.60 | 13.44 | 25.11 | 22.74 | - |

Notes: "-" means below detection limit.

**Table 4.** LA-ICP-MS in situ analysis of fluid inclusions in quartz of Dakalasu No. 1 pegmatite vein (ppm).

| Sample | A18 | A19 | A20 | A21 | A22 | A23 | A24 | A25 |
|---|---|---|---|---|---|---|---|---|
| Li | 866.7 | - | 362.3 | - | 371.5 | 110.7 | 322.6 | 279.1 |
| B | 2813 | 3028 | 1004 | 3153 | 1434 | 5119 | 3161 | 4191 |
| Na | 27,734 | 27,066 | 29,662 | 27,734 | 3096 | 28,364 | 27,066 | 263,967 |
| K | 1136 | 520.7 | - | 698.8 | 959.7 | 1575 | 692.1 | 710.7 |
| As | 518.5 | 845.1 | 963.2 | 1052 | 925.5 | 1133 | 726.3 | 1097 |
| Rb | 8.83 | 8.71 | 15.12 | 5.40 | - | 18.20 | 27.89 | 26.73 |
| Cs | 166.6 | 101.2 | - | 149.2 | - | 224.2 | 233.9 | 245.2 |
| Rb/Na | 0.000318 | 0.000322 | 0.00051 | 0.000195 | | 0.000642 | 0.00103 | 0.001013 |
| Cs/Na | 0.006007 | 0.00374 | 0.005378 | | | 0.007905 | 0.008641 | 0.009288 |
| Cs/Rb | 18.87 | 11.62 | | 27.62 | | 12.32 | 8.386 | 9.172 |

Notes: "-" means below detection limit.

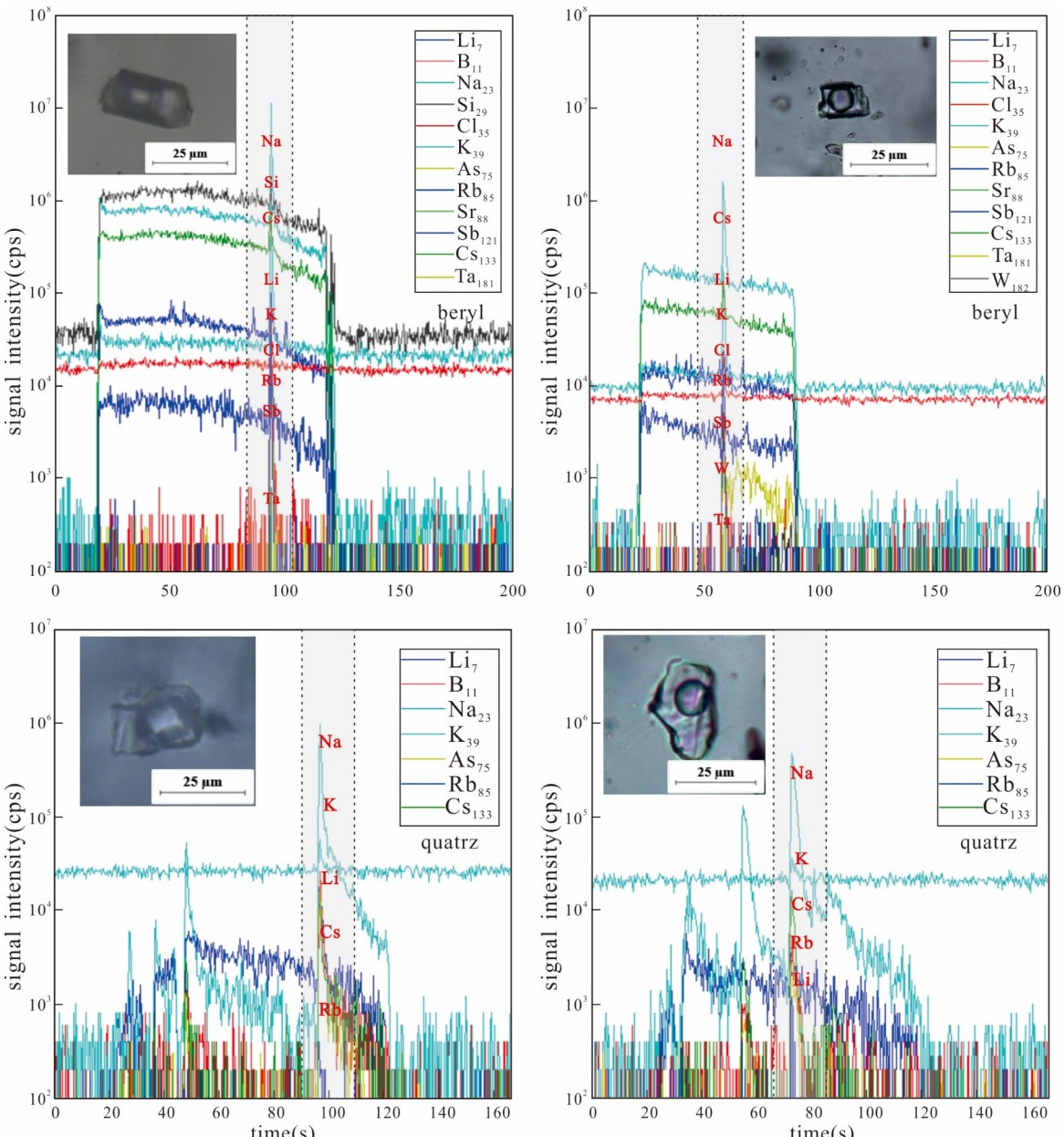

**Figure 10.** LA-ICP-MS time resolution profiles of single fluid inclusions in various minerals of Dakalasu No. 1 pegmatite vein.

The $CO_2$-bearing inclusions in the beryl of the rim zone show a certain amount of Li, B, K, Rb, etc. Among them, the highest contents of Li and B reached 2172 ppm and 34,407 ppm (absolute concentration with $\pm1\sigma$ uncertainty), respectively. The contents of Cs ranged from 669.6–3842 ppm. In contrast, the contents of Rb and Ta are lower, ranging from 54.74–322.9 ppm to 1.21–1.23 ppm, respectively.

Compared with the rim zone, Li, B, K, Rb, and Cs in the fluids of the core zone showed a decreasing trend, and their contents in some inclusions were below the detection limits. The highest content of Li was only 866.7 ppm. The concentrations of Rb and Cs were much lower at 8.83–27.89 ppm and 101.2–245.2 ppm, respectively.

In the fluid inclusions of rose quartz from transition zone, Li, B, Rb, and Cs contents are usually between those of beryl and quartz. The high Ti content detected in the rose quartz is the reason for the pink color of its crystals.

## 5. Discussion

### 5.1. Nature and Evolution of the Ore-Forming Fluids

The formation of pegmatite is an evolutionary process from magmatic to hydrothermal fluids. In terms of inclusions, a melt inclusion is captured in the magmatic stage, a fluid inclusion is recorded in the hydrothermal stage, and a fluid-melt inclusion is caught in the magmatic-hydrothermal transition stage [70]. Abundant $CO_2$-rich fluid inclusions and a few fluid-melt inclusions are developed in beryl and quartz in the rim and core zones of the Dakalasu No. 1 pegmatite vein. This suggests that the deposit has experienced a magmatic to hydrothermal growth evolution [71,72]. Microthermometry results show that the fluid in the rim have medium to high temperature (240~300 °C) and medium-high salinity (7.2%–10.3% $NaCl_{eqv}$) and is rich in volatile components such as $CO_2$ and $CH_4$. The homogeneous temperature (220–270 °C) and salinity (7.0%–8.8% $NaCl_{eqv}$) of $CO_2$-rich inclusions in the quartz of the core are slightly lower than those in the rim.

The LA-ICP-MS analysis of the individual inclusions shows that the fluids in the rim are enriched in Li, B, K, Zn, Rb, Sb, Cs, and As. In contrast, Li, B, K, Rb, and Cs concentrations are significantly lower in the core.

The detectable mineralized rare metal elements such as Nb and Ta in fluid inclusions are low, and only a trace amount of Ta (1.21–1.23 ppm) was detected in beryl. This indicates that Nb and Ta tend to concentrate in the residual silicate melt during the magmatic-hydrothermal evolution, which is consistent with the results of previous experimental geochemical studies [73,74].

The fluid temperature and salinity of the Dakalasu No. 1 pegmatite vein gradually decrease from the rim to the core, and the fluid composition changes accordingly. The mineral textural zonation can also indicate this feature. Earlier crystallized tantalum-niobium minerals, columbite and strueverite, are mainly distributed in the rim, while later, crystallized tapiolite and apatite are produced primarily in the core [35].

### 5.2. The Role of CO$_2$-Rich Fluids in Rare Metal Enrichment and Mineralization

As important mineralizing agents, volatile components, such as $CO_2$, B, and F, played a critical role in rare metal enrichment mineralization [75,76]. $CO_2$ significantly affects the depth of magma degassing and the properties of fluids (vapor and brine), which can lead to phase separation of the $NaCl-H_2O$ system under higher P-T conditions [77–79]. Furthermore, because of the strong depolymerization properties and affinity of $CO_2$, the diffusion rate and activity of rare element cations and volatiles in the melt are increased. This leads to the rapid separation of volatile-rich melts from the parent magma and enhances the occurrence of immiscibility [80]. As the pressure decreases, $CO_2$ escapes earlier than other volatile components, which causes a change in the pH of the mineralizing fluid and leads to rare metal precipitation [81,82].

The fluid inclusions in various textural zones of the Dakarlasu No. 1 pegmatite vein are characterized by the richness of volatile components such as $CO_2$, $CH_4$, and $N_2$. The calculated results show (Table 1) that the $CO_2$ density from rim to core decreased from 0.71~0.84 g/cm$^3$ to 0.65~0.78 g/cm$^3$. The LA-ICP-MS analysis results also showed that the content of Li, B, K, Rb, Sr, and Cs decreased from the rim to the core zone. This synergistic variation relationship (Figure 11) partially reflects the possible contribution of $CO_2$ to the migration and enrichment of these rare elements.

Suo et al. (2022) [38] performed LA-ICP-MS, and EPMA analysis on beryl from the Dakalasu No. 1 pegmatite vein, and the results showed high FeO contents and Na/Cs ratios of beryl. This indicates that the differentiation evolution of the pegmatite veins is low. The enrichment of Be is related to the crystallization of the melt at a highly undercooled state. The melt-fluid immiscibility promoted by volatile components such as $CO_2$ is likely to be the effective mechanism leading to the enrichment of rare metal elements such as Be-Nb-Ta into mineralization.

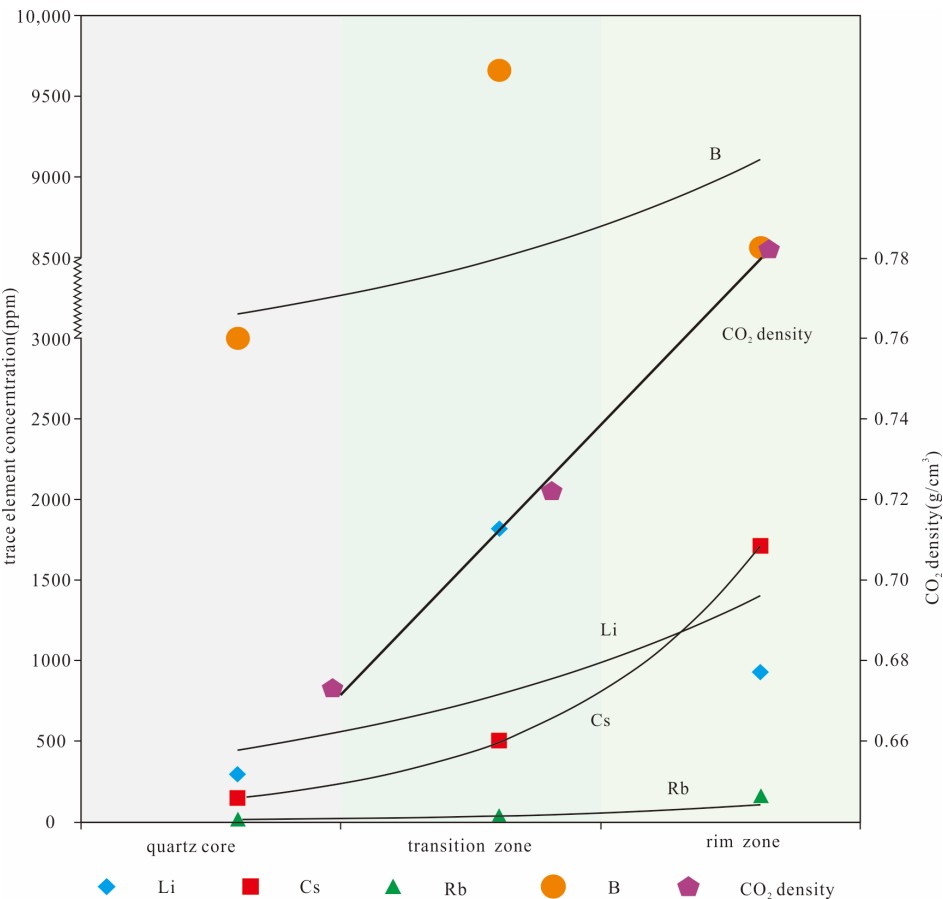

**Figure 11.** Covariance diagram of rare metal elements and $CO_2$ content in each textural zone.

## 6. Conclusions

(1) Numerous $CO_2$-rich fluid inclusions are developed in various mineral textual zones of the Dakalasu No. 1 pegmatite vein. The fluid composition is mainly a NaCl-$H_2O$-$CO_2$ system. The fluid temperature and salinity show a decreasing trend from the rim to the core.

(2) The LA-ICP-MS analysis of individual fluid inclusions reveals that the rim zone fluids are enriched in Li, B, K, Zn, Rb, Sb, Cs, and As, while the contents of these elements are significantly lower in the core.

(3) $CO_2$ may play a vital role in migrating and enriching rare metals such as Be, Nb, and Ta in the Dakalasu No. 1 pegmatite vein. The melt-fluid immiscibility that occurs in the system during the magmatic-hydrothermal transition stage may be an effective mechanism for rare metal elements' mineralization.

**Author Contributions:** J.Z. wrote the paper, S.G. developed research ideas and analysis plan, Y.T. experiment and data analysis, H.Z. and Z.L. modified the article. All authors have read and agreed to the published version of the manuscript.

**Funding:** This study is jointly supported by the Key Program of the National Natural Science Foundation of China (No. 91962222) and Guizhou Provincial 2019 Science and Technology Subsidies (No. GZ2019SIG).

**Institutional Review Board Statement:** Not applicable.

**Informed Consent Statement:** Not applicable.

**Data Availability Statement:** All data generated or used during the study appear in the submitted article.

**Acknowledgments:** We sincerely thank Xinjiang Non-ferrous Metals Group for their support for field work. We would also like to thank Junying Ding, Zhe Chi and Jianming Cui of Nanjing University for their experimental assistance. This manuscript benefited considerably from the comments and suggestions of the Editor and two anonymous reviewers.

**Conflicts of Interest:** The authors declare no conflict of interest.

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
