# Peer review of "Characteristics and Geological Significance of CO2-Rich Fluid Inclusions in Dakalasu No. 1 Pegmatite Dyke, Altay"

_minerals, doi:10.3390/min13030365_

Round 1
Reviewer 1 Report
A brief revision of the paper is needed. Some terms are not of all accepted by the relevant literature (e.g., most of the authors prefers "wall" or "rim" instead "edge" for describing the peripheral part of a pegmatite body). There are a lot of small editorial improvements to be do, but I preferred to suggests the change in the attached file. Abbreviations for minerals must be fitted with the recommendations of Whitney and Evans (2010) as suggested in the attached file. The paper is good and deserves publication.

Author Response
Response to Reviewer 1 Comments
Point 1: Some terms are not of all accepted by the relevant literature (e.g., most of the authors prefers "wall" or "rim" instead "edge" for describing the peripheral part of a pegmatite body).
Response 1: Thank you for noting this out. We have replaced “edge” with “rim” in the manuscript, including replacements in the text (line15\21\23...) and some of the figures (figure 7, 8 and 11).
Point 2: There are a lot of small editorial improvements to be do, but I preferred to suggests the change in the attached file.
Response 2: Thank you very much for pointing out these shortcomings. We have revisited each of them in the light of your comments.
Point 3: Abbreviations for minerals must be fitted with the recommendations of Whitney and Evans (2010) as suggested in the attached file.
Response 3: Thank you for the suggestions. All the mineral abbreviations in the manuscript have been modified according to Whitney and Evans (2010).
Point 4: The paper is good and deserves publication.
Response 4: Thank you very much for your encouragement and we will continue to work hard to publish more findings.

Reviewer 2 Report
In Figure 9 A and B the sharp peak at about 3600 cm-1 is not labeled or discussed. In the text there are a few places where the English could be improved, but it does not seem important. In the References some spaces between words have been left out, but again it does not seem important.
This manuscript describes laboratory studies on samples from one pegmatite dike. It does not describe field studies. It is not ever clear whether the authors themselves collected the samples studied. Therefore Figures 1, 2, and 3 and the call outs to them on lines 85, 90, and 96 seem unnecessary.
In lines 101 and 102 it is stated that the BeO content of the blocky pegmatite zone is as high as 0.49%. No indication is given as to how that was determined, by whom, or whether it is weight % or volume %. In the next line it is stated that beryl is 0.56% abundant in the whole vein. It is unnecessarily confusing to switch from beryl to the BeO content of the beryl in the 2 statements. In line 105 aplite is listed as a mineral, but it is not.
Section 4.4 describing the mass spectrometry seems far too short to really explain what was done. Are the numbers listed in the tables mass fractions or atom fractions? How were they obtained from the data? What are the uncertainties?
Now I will mention minor errors, most probably caused by type setting.
Line 25 focused should be ranged. Line 148 MS should be Ms. Line 255 4.3 “Raman Spectroscopy” is in wrong place. Line 339 NO should be No. Line 316 “aplite” is incorrect. Figure 9 sharp line at about 3600 cm-1 should be explained. Figure 10 samples should be identified and x-axis should be labeled. Table 4 next to last 2 rows are misaligned. There are minor typos in References 18, 22, 41, 69,and 80.
Author Response
Response to Reviewer 2 Comments
Point 1: This manuscript does not describe field studies. It is not ever clear whether the authors themselves collected the samples studied. Therefore Figures 1, 2, and 3 and the call outs to them on lines 85, 90, and 96 seem unnecessary.
Response 1: Thank you. The second part of the manuscript introduced the general geological background of the research area. We believe that information on the geological background is beneficial to the readers in gaining a clearer understanding of the basic characteristics of the deposit. Detailed field work was carried out in the study area and relevant samples were systematically collected for further study.
Point 2: In lines 101 and 102 it is stated that the BeO content of the blocky pegmatite zone is as high as 0.49%. No indication is given as to how that was determined, by whom, or whether it is weight % or volume %. In the next line it is stated that beryl is 0.56% abundant in the whole vein. It is unnecessarily confusing to switch from beryl to the BeO content of the beryl in the 2 statements. In line 105 aplite is listed as a mineral, but it is not.
Response 2: Thank you for noting this out. The content mentioned in lines 101 and 102 is the volume percentage of beryl. These data are quoted from [35](Zou, 2006), and we have added citations in the corresponding positions of the manuscript. In addition, to avoid confusion, we have consistently referred to BeO and beryl as beryl. In line 105, ‘apatite’ was misspelled as ‘aplite’. We have made the changes, thanks again.
Point 3: Section 4.4 describing the mass spectrometry seems far too short to really explain what was done. Are the numbers listed in the tables mass fractions or atom fractions? How were they obtained from the data? What are the uncertainties?
Response 3: Thank you for your suggestions. The data we reported here are absolute concentrations (±1σ uncertainty) of elements in individual fluid inclusions. They were calculated from the element ratios determined by LA-ICP-MS. NaCl equivalent salinities obtained from microthermometry were used as internal standards. References [64]-[65] provide a detailed description of the measurement and data processing procedure, which we have cited in section 3.3 of the manuscript.
Point 4: Line 25 focused should be ranged. Line 148 MS should be Ms. Line 255 4.3 “Raman Spectroscopy” is in wrong place. Line 339 NO should be No. Line 316 “aplite” is incorrect. Figure 9 sharp line at about 3600 cm-1 should be explained. Figure 10 samples should be identified and x-axis should be labeled. Table 4 next to last 2 rows are misaligned. There are minor typos in References 18, 22, 41, 69,and 80.
Response 4: Thank you for noticing these issues. We have carefully revised them one by one.
Thank you again for your constructive comments and valuable time.
